# A Genome-Wide Search for Candidate Genes of Meat Production in Jalgin Merino Considering Known Productivity Genes

**DOI:** 10.3390/genes13081337

**Published:** 2022-07-26

**Authors:** Alexander Krivoruchko, Alexander Surov, Antonina Skokova, Anastasiya Kanibolotskaya, Tatiana Saprikina, Maxim Kukharuk, Olesya Yatsyk

**Affiliations:** 1FSBSI North Caucasian Federal Scientific Agrarian Center, 356241 Mikhailovsk, Russia; surov.stv@yandex.ru (A.S.); antoninaskokova@mail.ru (A.S.); dorohin.2012@inbox.ru (A.K.); saprikina.tanya@mail.ru (T.S.); malteze@mail.ru (O.Y.); 2Federal State Autonomous Educational Institution of Higher Education, North Caucasian Federal University, 355017 Stavropol, Russia; kuth87@mail.ru

**Keywords:** sheep, SNP, genome-wide association search, GWAS, candidate gene, Jalgin merino

## Abstract

In a group of Jalgin merino rams with no significant influence on the dispersion of the phenotypes of known productivity genes (*MSTN*, *MEF2B*, *FABP4*, etc.), a genome-wide search for associations of individual polymorphisms with intravital indicators of meat productivity was performed. Using the Ovine Infinium HD BeadChip 600K, 606,000 genome loci were evaluated. Twenty-three substitutions were found to be significantly associated with external measurements of the body and ultrasonic parameters. This made it possible to describe 14 candidate genes, the structural features of which can cause changes in animal phenotypes. No closely spaced genes were found for two substitutions. The identified polymorphisms were found in the exons, introns, and adjacent regions of the following genes and transcripts: *CDCA2*, *ENSOARG00000014477*, *C4BPA*, *RIPOR2*, *ENSOARG00000007198*, *ENSOARG00000026965 (LincRNA)*, *ENSOARG00000026436 (LincRNA)*, *ENSOARG00000026782 (LincRNA)*, *TENM3*, *RTL8A*, *MOSPD1*, *RTL8C*, *RIMS2*, and *P4HA3*. The detected genes affect the metabolic pathways of cell differentiation and proliferation and are associated with the regulation of the immune system. This confirms their possible participation in the formation of the phenotypes of productivity parameters in animals and indicates the need for further study of the structure of candidate genes in order to identify their internal polymorphisms.

## 1. Introduction

Genomic and marker-associated breeding methods are becoming an integral part of modern animal husbandry. They allow one to effectively assess the breeding value of animals and predict productive qualities. This significantly accelerates the improvement of the used breeds and the breeding of new ones. However, the further development of genetic technologies in animal husbandry requires more information about the genes that affect the phenotypes of economically valuable traits [1].

The intensity of the growth and development of muscle tissue in animals is the basis of their meat productivity. They depend both on the rate of synthesis of the proteins of muscle cells and on the action of various growth factors, regulators of intercellular interactions, hormones, etc. This whole system is regulated by a large number of genes combined into gene networks [2]. Previous studies have identified a number of key genes that have the strongest impact on muscle development. One of the first to be described was the myostatin gene (*MSTN*, *GDF-8*), which limits the growth of muscle tissue. The appearance of mutations in the gene region that reduce its functional activity leads to increased muscle growth and the formation of a double muscle phenotype [3,4]. The increased expression of the follistatin (*FST*) gene stimulates muscle growth due to the ability of the protein product to inhibit the myostatin protein [5]. The *MYOD1* gene regulates the expression of the myostatin gene and, through it, affects meat productivity [6,7]. Mutations at the callipyge locus are associated with increased hip muscles in animals [8].

As a result of the ongoing selection process to improve the meat qualities of animals, some mutations in these genes are fixed in the form of homozygotes and cease to affect the diversity of the phenotype. This is clearly seen in the example of the c.* 1232G > A (HGVS nomenclature) substitution in the myostatin gene. In the vast majority of the meat breeds Texel and White Suffolk, the homozygous allele A can be found, which is associated with the microRNA-mediated inhibition of myostatin gene translation and the development of muscle hypertrophy [9,10]. In Merino sheep, on the contrary, the G allele is fixed in the homozygous variant. Thus, the substitution cannot be used as a marker in breeding for improving meat productivity in these breeds [11].

The effect of protein products of many genes on muscle tissue is not as pronounced as in those listed above. Therefore, their influence is often not detected in studies on the association of polymorphisms in genomes with productive traits. However, in breeds adapted to special environmental conditions (for example, climate) and a food supply, they can show their effect. This should be especially noticeable in individuals with a similar genotype for known meat productivity genes, which does not significantly affect the diversity of exterior and interior indicators [12].

A promising breed for identifying associations of new genome loci with meat productivity parameters is the Jalgin merino sheep. This breed of sheep was bred in the Stavropol Territory and registered in 2013. Sheep of the Stavropol breed and Australian merinos served as the basis for it. The breed turned out to be perfectly adapted for breeding in arid steppes with a high salt content in the soil, which tolerates summer heat and cold winters well. Along with high-quality wool with a fineness of up to 19 microns, it is distinguished by a satisfactory meat form. Rams reach a weight of more than 120 kg, and ewes more than 50 kg. At the age of one year, rams have a weight of up to 80 kg, and ewes almost 40 kg. The presence of sufficient variability in meat productivity indicators in the breed makes it promising for breeding in terms of improving meat yield parameters [13].

The aim of our work was a genome-wide search for associations of individual genome loci with lifetime indicators of meat productivity and new candidate genes in Jalgin merino sheep. We also used a new approach based on the choice of animals for study, in which there are no significant associations of polymorphisms of known productivity genes with phenotype parameters.

## 2. Materials and Methods

### 2.1. Animals Used and Obtaining Samples

Ethics statement. The sample collection and study purposed were approved by the Institutional Animal Care and Use Committee (approval number 2021-0043, 11 October 2021) of the All-Russian Research Institute of Sheep and Goat Breeding, Stavropol, Russian Federation.

Rams of the highest-quality of the Jalgin merino breed (Figure 1) at the age of 12 months were the object of this study. Among 224 rams of one herd of the breeding core in the Stavropol Territory (Russian Federation), 50 rams with maximal pure breed quality were genotyped. To confirm the purity of the breed, we used genotyping data for 50 rams of the Manych merino and 50 rams of the Stavropol breed. In selected Jalgin rams, the lifetime parameters of meat productivity were determined. Live weight at birth and at 12 months was measured using scales, and daily weight gain was calculated from these parameters. The height at wither and croup, the width of the chest and back, the depth of the chest, and the girth of the shoulder, forearm, and thigh were measured with a grading ruler and measuring tape [14]. The parameters of the depth and width of the muscle «eye», the depth of the femoral muscle, and the adipose tissue in the lumbar region were determined via ultrasound [15]. All animals were clinically healthy, kept in optimal conditions, and fed with a total mixed ration.

To confirm the purity of the Jalgin merino breed, a comparative analysis of genotypes was performed with the two main breeds most common in the Stavropol Territory: the Manych merino and the Stavropol breed (Figure 2). In the space of the first two principal components, these breeds were clearly clustered into three groups, without significant intersections with each other.

### 2.2. Extraction of DNA and Genotyping

Genomic DNA was isolated from whole blood samples taken under aseptic conditions from the jugular vein using the Pure Link Genomic DNA MiniKit (Invitrogen Life Technologies, Carlsbad, CA, USA) in accordance with the manufacturer’s protocol. DNA was isolated from 100 µL of blood. The average concentration was 25 ng/µL. Animal genotyping was performed using the Ovine Infinium HD BeadChip 600K (Illumina Inc., San Diego, CA, USA) according to the manufacturer’s protocol. Genotyping was carried out in the laboratory of the Skolkovo Institute of Science and Technology SCOLTECH (Moscow, Russian Federation). The initial processing of the genotyping results was performed using the Genome Studio 2.0 software (Illumina Inc., San Diego, CA, USA).

### 2.3. Quality Control of Genotyping

Quality control of genotyping was carried out using PLINK V.1.07 software [16]. The data processing included samples with a call rate of detected SNPs of more than 0.95. Substitutions with a minor allele frequency (MAF) of less than 0.05 and a missing genotype of more than 0.1 were also excluded. The value *p* = 0.0001 was used as the threshold according to the Hardy–Weinberg equilibrium criterion (HWE). Among the 606,006 SNPs, 39,811 failed the frequency test, and 834 markers were excluded based on the HWE test. Additionally, 12,804 SNPs failed the missingness test. With a positive result, 50 samples of animals studied in the second stage underwent genotyping quality control. After removing 432 SNPs with unknown positions (chromosome 0), 560,381 SNPs were used for further analysis.

### 2.4. Genetic and Statistical Analysis

The genome-wide association study was performed using the PLINK V.1.07 software. It was based on the assessment of the significance of SNP influence on quantitative meat production trait variability. SNPs on the X chromosome were not excluded, since the PLINK software allows for one to search for associations in males, taking this into account also when controlling the quality of genotyping. Visualization and graphing were performed using the “QQman” package in the programming language “R”. The principal component analysis was performed using the “SNPRelate” package in the programming language “R”. Heatmaps were designed in the Genome Studio 2.0 software (Illumina Inc., San Diego, CA, USA). The search for candidate genes was performed in half of a centimorgans region (250,000 bp upstream and downstream) around the SNPs that showed significant associations with meat production traits. For SNP mapping and alignment, the Ovis_Aries_3.1 genome assembly was used. Gene annotations were performed using the genomic browsers UCSC (www.genome.ucsc.edu, accessed on 20 June 2022) and Ensembl (www.ensembl.org, accessed on 20 June 2022). For an analysis of productive parameter differences between animals with some genotypes, Student’s t-criterium in Excel software (Microsoft, Redmond, WA, USA) was used. A significant difference was detected if *p*-value < 0.01. Gene ontology (GO) analysis was performed using g:Profiler (version e106_eg53_p16_65fcd97, database updated on 18 May 2022). Sets of more than 5 and less than 500 genes were selected for testing. The results were considered significant at *p* < 0.05.

## 3. Results

At the first stage of the research, we established the genotypes of the Jalgin merino rams through polymorphisms in the loci of known meat productivity genes. Based on the data obtained, a group of animals was formed with the closest genotype for 19 SNPs located in the region of eight marker genes with a proven effect on the yield of meat products. Table 1 shows the results of the associative analysis of these SNPs in a group of selected animals with two in vivo phenotype parameters directly related to the level of meat productivity: live weight and muscle «eye» depth. The considered SNPs did not have a significant relationship with the live weight index. For the substitutions rs408627462 and rs416984116, located in the region of the *GH* and *MYOD1* genes, respectively, the level of significance of *p*-value associations with the muscle «eye» depth was 0.04. This was a low value, but it was still higher than our threshold of significance of differences (*p*-value < 0.01).

A heat map of the distribution of polymorphisms in genes described as affecting meat productivity demonstrated the absence of any regularity in the distribution of genotypes for these genes in the studied animals (Figure 3). According to the results of genotyping, in Jalgin merino, the analyzed loci in the *MSTN* and *FABP4* genes, as well as one locus in the *MEF2B* gene region, were monomorphic. The remaining SNPs were represented by different genotypes that did not have a visible pattern of distribution between individuals. An attempt to cluster by identified genotypes was unsuccessful. Clustering showed the presence of a large number of groups of genotypes. Several genotypes were clustered as single representatives of their class.

In general, the presented picture characterizes the absence of the dependence of the variance of phenotypic indicators in the selected group of Jalgin merino rams on both individual SNPs and combinations of polymorphisms into genotypes.

A genome-wide search for associations of individual SNPs with in vivo productivity parameters revealed a number of substitutions that showed significant signs of association (Figure 4). Thus, the confidence threshold at −log_10_(*p*) = 5 in our study of associations with chest depth was overcome by more than 10 SNPs. The highest significance values were for the substitution on chromosome 20. These parameters were slightly lower for substitutions on chromosomes 2 and 12. Most SNPs with significance values above the threshold value were identified for back width. Among them, substitutions on chromosomes 16, 18, and 26 had the maximum values. Associations with muscle «eye» width greater than the threshold had one substitution on chromosome 16 and a whole pool of SNPs on chromosome X, located within the same locus. For back width, associations with the highest significance values were found on chromosomes 9 and 15. In addition, there were more than 10 substitutions that passed the threshold of the significance parameter. For other measurements of in vivo indicators of meat productivity in Jalgin merino, we did not reveal significant associations with genome loci.

To determine the potential candidate genes that affected the productive qualities of sheep, we selected 23 SNPs with the highest −log_10_(*p*) values. They were used to annotate the genes containing the detected SNPs or located next to them within half of a centimorgan (Table 2). The study showed that most of the SNPs associated with phenotype parameters in sheep were located on chromosomes 2 and X. There was only one polymorphism in gene exons, 10 SNPs were localized in introns, and one substitution was located in the 3′ downstream region. Three substitutions were associated with long noncoding RNA. Two substitutions on chromosome 2 were identified in the intergenic space and were too far from the nearest genes, outside the search interval for candidate genes that we determined.

Based on the totality of the SNPs we found with the most reliable relationship with lifetime productivity indicators in Jalgin merino, complex genotypes were compiled and analyzed (Figure 5). The performed clustering of genotypes made it possible to identify two large main groups of the third order on the heat map, which included 45 of the 50 animals studied. The remaining five individuals were clustered into three separate groups: one individual in a second-order cluster and two groups of two individuals of the third order. For the bulk of the examined animals, clustering into two groups was performed based on five polymorphisms associated with four candidate genes: *RTL8A*, *RTL8C*, *ENSOARG00000007198*, and *MOSPD1*. The “Genotype A” group included 30 animals with a mutant homozygous genotype for the *RTL8A* gene, and the “Genotype B” group included 15 individuals with a wild homozygous genotype for this gene. Two polymorphisms were associated with the same *RTL8A* gene. Three of these four genes were located on the X chromosome and were associated with the width of the muscle «eye».

An analysis was performed on the significance of differences between groups of animals with different genotypes according to the studied parameters of the phenotypes (Table 3). Genotype A carriers were found to be significantly different from genotype B animals in terms of body weight at one year of age, daily weight gain, and muscle «eye» depth and width. The live weight of genotype A was 7% higher, and daily weight gain was 8% higher. In relation to the muscle «eye», an inverse relationship was observed. Its dimensions were larger in animals with genotype B—width by 9% and depth by 2.5%. No significant differences were found among other parameters.

In our study, the results of a genome-wide search for candidate genes of productive traits were supplemented by an ontology analysis of genes and metabolic pathways that may affect the manifestation of phenotypic traits. As a result, it was found that out of 14 potential candidate genes in the GO analysis, 3 Linc RNAs and 2 uncharacterized sequences did not pass. For the remaining candidate genes, 15 biological processes and 5 molecular functions were significantly identified (*p* < 0.05) (Figure 6). Molecular functions have been associated with proline metabolism and ascorbic acid transport. Biological processes included regulatory pathways for the development and operation of neurons, the morphogenesis of sensory organs, and the regulation of neutrophil functions. A number of metabolic pathways have been associated with protein metabolism, guanyl biotransformation, and GTP.

## 4. Discussion

We used a new approach in a genome-wide search for candidate genes of productive traits in Jalgin merino. It consisted of the selection of an experimental group of animals in which, according to the results of an analysis of associations, polymorphisms of known marker genes (*MSTN*, *MEF2B*, *FABP4*, etc.) had no significant effect on lifetime parameters of meat productivity. In our opinion, this was due to either the presence of these genes in all animals in the homozygous variant or to the presence of a combination of one allele in the homozygous variant with heterozygotes. In the first case, the absence of a difference in genotypes neutralized the effect on the phenotype dispersion; in the second case, the presence of only one allelic variant of a gene in the genome was probably not enough to realize visible changes in the phenotype. This may be due to the intermediate type of inheritance of traits or the incomplete dominance of allelic variants. In addition, the absence of a pronounced relationship between the polymorphism of genes that control meat productivity and phenotypic traits may be due to the epistatic or modifying interaction of genes, which was confirmed by the absence of genotype clustering in more or less large groups.

According to our assumptions, the dispersion of the value of quantitative traits of productivity in the group of animals we selected depended on the genes, and the impact on productivity, which was not previously described, manifested itself only as a result of the stabilization of the impact of known genes after a long selection process.

The analysis of genotypes based on the method of principal components confirmed the purity of our sample of animals of the Jalgin merino breed. Two merino breeds (Jalgin and Manych merinos) clustered among themselves only in the second component, which indicated a fairly close origin. The Stavropol breed was clustered completely separately in both components, being located in the first component between the Merino breeds. This was consistent with the history of the origin of the studied merinos, since the sheep of the Stavropol breed were among their common ancestors.

A genome-wide search for SNP associations with lifetime indicators of meat productivity in the group of animals we examined showed the presence of a relationship between trait variability and genome loci in four parameters: three external measurements and one determined with ultrasound. The latter could be considered the most accurate in assessing meat productivity, since it directly characterized the width of the lumbar muscle. The loci found in other studies did not show associations with meat productivity parameters. The genes in which the SNPs we identified were localized were actively involved in the regulation of cell proliferation and differentiation. This indicated their involvement in the implementation of genetic programs for the growth and development of animals.

The rs403409170 substitution was located on chromosome 2 in exon 12 of the *CDCA2* gene. It encoded cell division cycle-associated protein 2. As a result of the replacement of guanine with adenine, the missense mutation changed the structure of the GCA codon to ACA. In the amino acid chain, at position 539, alanine changed to tryptophan. Cell division cycle-associated protein 2, also called RepoMan, is a regulatory subunit of key cell cycle phosphatases [17]. The CDCA2 enzyme is involved in the dephosphorylation of histone H3 during the mitotic cycle. This shows its role in chromatin remodeling in the interphase nucleus, where it also provides a response to DNA damage of various origins [18]. An increased expression of the *CDCA2* gene in humans has been found in malignant neoplasms [19]. Studies of the effect of the CDCA2 protein on the growth and development of animals have not been previously conducted. However, its important role in cell proliferation and the detection of changes in the amino acid sequence as a result of the presence of SNPs make it possible to consider *CDCA2* as a promising gene candidate for productivity.

At a distance of approximately 100 kbp from the SNP rs417405143, there is the *ENSOARG00000014477* gene encoding the J-domain-containing protein. The group of proteins containing the J-domain and with a mass of about 40 kDa belongs to “heat shock” proteins. They ensure the functioning of the chaperone complex for amino acid chain folding, a function in membrane protein transport complexes, and control the activity and stability of various proteins [20,21]. Participation in a number of intracellular processes that play an important role in metabolism allowed us to consider the *ENSOARG00000014477* gene as a candidate associated with animal growth.

The substitutions rs412292790 and rs398401406, which have a significant relationship with the depth of the chest, are located in intergenic space. Within half of a centimorgans from them, we did not find coding DNA regions. It is possible that these SNPs are linked to genes located almost 300 kbp apart. However, the criterion we chose for searching for candidate genes did not allow us to include such distant genes into their composition. However, the SNPs we identified could be used as independent molecular genetic markers in the selection of sheep of the Jalgin merino breed.

The *C4BPA* (*complement component 4 binding protein* α) gene contains the rs421040859 substitution in introns 9–10, which also showed an association with chest depth in the examined animals. This gene encodes the α chain (there are seven of them in the protein) of a soluble plasma inhibitory glycoprotein that protects body tissues from the autoimmune reaction of the complement system. The gene is a highly conserved and phylogenetically ancient component of the animal immune system [22]. Mutations in the gene in humans lead to spontaneous abortion [23] and atypical hemolytic uremic syndrome [24]. The importance of the physiological functions of the C4BPA gene allowed us to consider it as a candidate gene for animal productivity parameters.

The rs414911966 substitution located on chromosome 20 is located in the intron of the *RIPOR2* gene. It encodes a protein from the RHO-family-interacting cell polarization regulator 2. The biological function of this regulator is to provide the mechanisms of cell polarization and migration, including the implementation of the body’s immune response [25]. High expression of the *RIPOR2* gene in hair follicle cells has been associated with the regulation of hair growth, which has been confirmed by the presence of polymorphisms that cause hair loss in humans [26]. Disorders in the *RIPOR2* gene are also accompanied by the development of oncological diseases [27]. The effect of the gene on productivity in animals has not been studied, but the described effects of mutations in it allowed us to consider it as a potential candidate gene for further study.

The intron of the *TENM3* gene contains the SNP rs429422002 that we identified. This gene encodes the teneurin-3 protein, which is highly expressed in hippocampal neurons. It belongs to cell adhesion molecules and coordinates the normal interaction of neurons during the laying of the hippocampus. In addition, the *TENM3* gene can form extracellular epidermal growth factor-like (EGF-like) as a result of alternative splicing [28]. Teneurin-3 is also required for the normal formation of connections between retinal ganglion cells, which ensures the development of the visual system in vertebrates [29]. The importance of the gene for the normal development of the nervous system allows us to classify it as a candidate gene for productivity.

Two substitutions, rs161648030 and rs427877945, are located next to the gene encoding retrotransposon Gag-like protein 8 (*RTL8A*). The rs415654848 substitution is located in the untranslated 3′ downstream region of the retrotransposon Gag-like protein 8C (*RTL8C*) gene. The genes of this family make up a significant part of the animal genome and belong to the evolutionarily ancient components that originate from retroviruses [30]. Due to their intragenomic mobility, retrotransposons can be a source of the formation of new genes or change the code of existing genes, affecting the functions of transcribed proteins. This is one of the adaptation mechanisms of living organisms during the evolutionary process [31]. We considered it possible to identify specific polymorphisms in the *RTL8A* and *RTL8C* genes that affect the productive qualities of animals.

Another substitution we found on the X chromosome (rs413531430) is located next to the *MOSPD1* gene. Its product belongs to proteins containing the motile sperm domain containing 1. In the intracellular metabolism, MOSPD1 provides contact interaction between the membranes of individual components of the endoplasmic reticulum and regulates the transport of large molecules [32]. Studies have shown its importance in the regulation of proliferation and the differentiation of mesenchymal stem cells [33]. Based on this, we attributed this gene to candidates that regulate the parameters of the animal phenotype.

The *RIMS2* gene, in the intron of which we found the rs406848373 substitution associated with back width, encodes the regulating synaptic membrane exocytosis 2 protein. It is located on the presynaptic membrane and belongs to the regulators of the release of neurotransmitters, on which the transmission of a nerve impulse depends. Polymorphisms in this gene have been found to be associated in humans with the development of degenerative lumbar scoliosis [34]. In [35], the authors revealed an association of mutations in the *RIMS2* gene with the development of neurodegenerative disorders and pancreatic damage, accompanied by a decrease in insulin secretion. These facts indicate the need for further study on the relationship between *RIMS2* gene polymorphisms and animal productivity indicators.

The rs427196452 substitution associated with the back width in the examined animals is located in the intron of the *P4HA3* gene encoding prolyl 4-hydroxylase subunit α 3. This protein is a component of the enzyme synthesizing collagen and participating in its folding. Due to its participation in the epithelial–mesenchymal transport of substances, it affects cell differentiation and can cause the development of malignant neoplasms [36]. The functional role of *P4HA3* makes it possible to attribute it to possible candidate genes for productive traits.

The genotypes constructed according to the profiles of the polymorphisms we identified clustered into two large subgroups. The animals belonging to them showed a significant difference in the three most important indicators of meat productivity: live weight, the daily gain associated with it, and the size of the muscle «eye». This once again proved the role of the candidate genes we proposed in the formation of the productive qualities of animals. In addition, several polymorphisms, based on the presence of which a clear clustering was performed in the genotype, are of interest as molecular genetic markers for identifying promising animals in the breeding process.

From the five molecular functions discovered during the GO analysis, four were associated with the activity of proline dioxygenase, also described as prolyl hydroxylase. This enzyme is widely represented in various organs and tissues, as it takes part in the modification of one of the most common connective tissue proteins: collagen. With its participation, the correct tertiary structure of the protein is formed and its functional activity is ensured [37]. The function associated with the transport of ascorbic acid provides the body with a natural antioxidant that prevents damage to proteins and DNA. In addition, ascorbic acid is an important cofactor in the synthesis of collagen and cholesterol, and it is involved in the metabolism of xenobiotics. A sufficient intake of ascorbic acid in the animal’s body is necessary for its normal growth and development [38]. Taking into account the important role of collagen and ascorbic acid in the growth and development of animals, we considered the involvement of the candidate genes we identified in the implementation of this molecular function as confirmation of their influence on the productive qualities of Jalgin merino. Many biological processes are associated with the development and functions of neurons, including the morphogenesis of sensory organs. Thus, they ensure the normal functioning of the nervous system of the animal, which is also responsible for the formation of normal eating behavior. This depends on providing the body with the nutrients necessary for growth and development [39]. The involvement of genes associated with productive traits in some neuron metabolic pathways has also been found by other researchers [40]. Biological processes include the regulation of guanyl-nucleotide exchange factor activity and GTP binding. They are united by their influence on the transmembrane transport of nutrients and ensure the response of the cell to external influences [41]. Biological processes associated with the functions of neutrophils are part of the normal functioning of the immune system. The resistance of the body to infections and parasitic invasions ensures a normal level of metabolism in animals, which is necessary for their growth and development [42]. Thus, the pathways we identified affected, in one way or another, animal phenotypes and contributed to the realization of productive qualities in Jalgin merino.

## 5. Conclusions

We performed a genome-wide search for associations of individual polymorphisms with intravital indicators of meat productivity in a group of rams of the Jalgin merino breed with a significant absence of influence on the dispersion of the phenotypes of known productivity genes (*MSTN*, *MEF2B*, *FABP4*, etc.). For genotyping, we used the Ovine Infinium HD BeadChip 600K, which allowed us to evaluate 606,000 genome loci. As a result of the research, 23 substitutions were found that were significantly associated with the exterior and ultrasonic parameters characterizing the size of the muscles. Based on these data, 14 candidate genes were annotated, the structural features of which could cause changes in the animal phenotype. The identified polymorphisms were found in the exons, introns, and adjacent regions of the following genes and transcripts: *CDCA2*, *ENSOARG00000014477*, *C4BPA*, *RIPOR2*, *ENSOARG00000007198*, *ENSOARG00000026965 (LincRNA)*, *ENSOARG00000026436 (LincRNA)*, *ENSOARG00000026782 (LincRNA)*, *TENM3*, *RTL8A*, *MOSPD1*, *RTL8C*, *RIMS2*, and *P4HA3*. The candidate genes we proposed affected the differentiation and proliferation of cells and were associated with the regulation of the function of the immune and nervous systems. This confirmed their possible participation in the formation of the phenotypes of productivity parameters in animals and indicated the need for further study of the structure of candidate genes in order to identify their internal polymorphisms.

## Figures and Tables

**Figure 1 genes-13-01337-f001:**
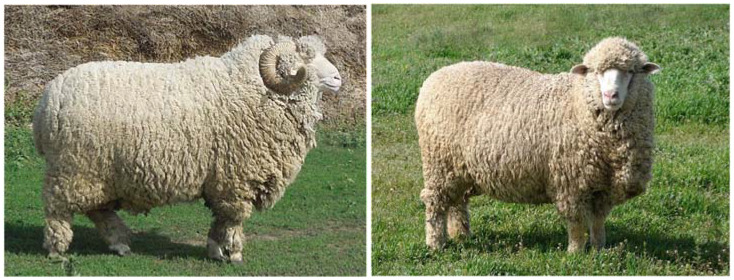
Ram (**left**) and ewe (**right**) of the Jalgin merino.

**Figure 2 genes-13-01337-f002:**
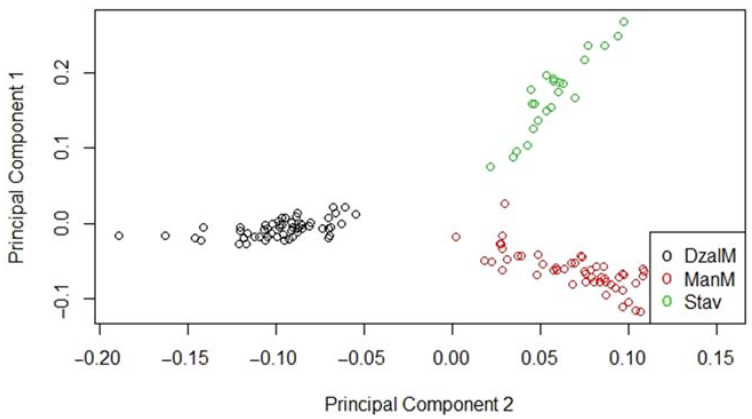
Breed purity PCA of Jalgin merino (DzalM) compared with the Manych merino (ManM) and Stavropol sheep (Stav).

**Figure 3 genes-13-01337-f003:**
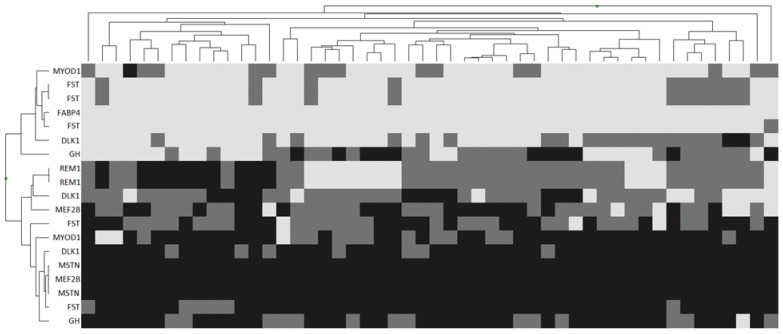
Heatmap of no association with meat production trait gene polymorphisms in Jalgin merino. Clustered individual genotypes. Light grey: wild homozygote; grey: heterozygote; dark grey: mutant homozygote.

**Figure 4 genes-13-01337-f004:**
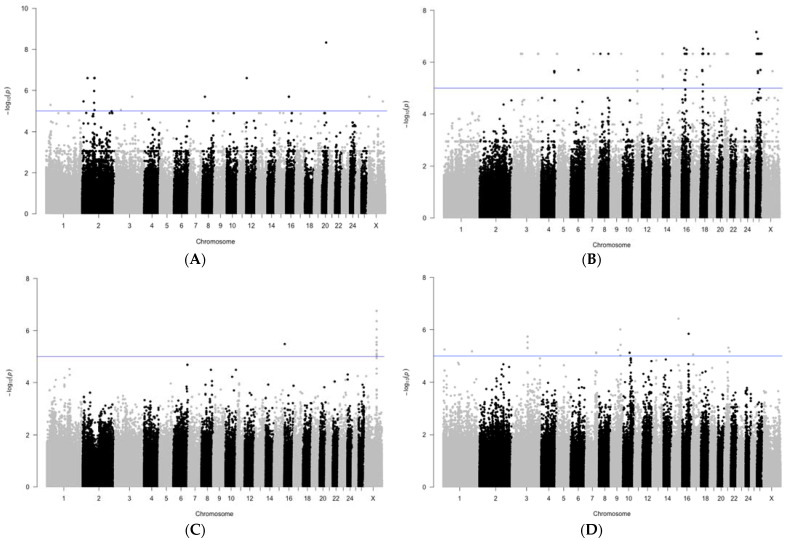
Manhattan plots of GWAS results with a set of −log_10_ (*p*) values for the studied SNP associations with meat production traits. Line indicates the threshold of significance of differences with a value of −log_10_ (*p*) = 5. (**A**) depth of chest; (**B**) width of chest; (**C**) width of muscle «eye»; (**D**) width of back.

**Figure 5 genes-13-01337-f005:**
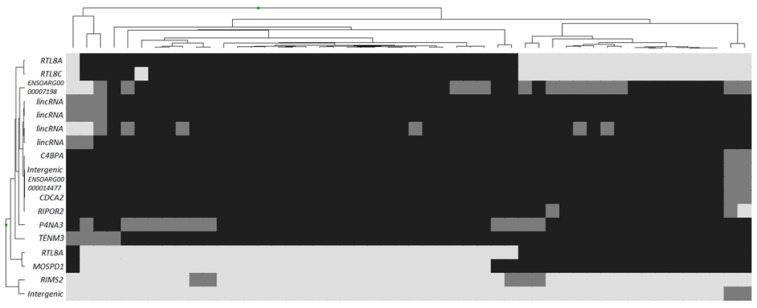
Heatmap of genotypes associated with meat production trait gene polymorphisms in Jalgin merino. Clustered individual genotypes. Light grey: wild homozygote; grey: heterozygote; dark grey: mutant homozygote.

**Figure 6 genes-13-01337-f006:**
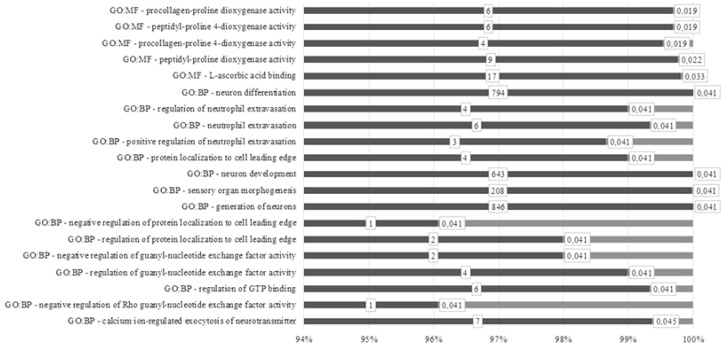
Gene ontology (GO) analysis for discovered candidate genes in Jalgin merino. GO:MF: molecular function; GO:BP: biological process. Light grey: adjusted *p*-value; dark grey: term size.

**Table 1 genes-13-01337-t001:** SNPs in genes with no associations of meat production parameters in Jalgin merino.

Gene	SNP	Chr	Position	SNP	LW, *p*-Value	DME, *p*-Value
*MSTN*	oar3_OAR2_118149265	2	118149265	rs868996529	N/A	N/A
*MSTN*	oar3_OAR2_118150665	2	118150665	rs408469734	N/A	N/A
*MEF2B*	oar3_OAR5_3860373	5	3860373	rs160009954	0.27	0.25
*MEF2B*	oar3_OAR5_3867887	5	3867887	rs406375431	N/A	N/A
*FABP4*	oar3_OAR9_57537070	9	57537070	rs428401213	N/A	N/A
*GH*	oar3_OAR11_47529756	11	47529756	rs410323075	0.93	0.93
*GH*	oar3_OAR11_47545769	11	47545769	rs408627462	0.46	0.04
*REM1*	oar3_OAR13_60384593	13	60384593	rs429901478	0.24	0.99
*REM1*	oar3_OAR13_60385591	13	60385591	rs160597933	0.24	0.99
*MYOD1*	oar3_OAR15_3434222	15	3434222	rs416984116	0.92	0.04
*MYOD1*	oar3_OAR15_3441596	15	3441596	rs424017339	0.14	0.64
*FST*	OAR16_27849538.1	16	25631318	rs412928257	0.74	0.72
*FST*	oar3_OAR16_25632659	16	25632659	rs161143581	0.93	0.43
*FST*	oar3_OAR16_25632701	16	25632701	rs162152758	0.25	0.34
*FST*	oar3_OAR16_25633632	16	25633632	rs401759231	0.93	0.43
*FST*	oar3_OAR16_25638968	16	25638968	rs399590017	0.41	0.33
*DLK1*	oar3_OAR18_64313560	18	64313560	rs403190653	0.21	0.24
*DLK1*	oar3_OAR18_64314938	18	64314938	rs403024346	0.88	0.52
*DLK1*	oar3_OAR18_64341672	18	64341672	rs10721508	0.94	0.12

Chr: chromosome; N/A: no SNPs detected; LW: live weight; DME: depth of muscle «eye».

**Table 2 genes-13-01337-t002:** Associated with meat production trait SNPs and candidate genes in Jalgin merino.

Trait	Chr	SNP	Position	*p*-Value	Alleles	Gene/Distance
DC	2	rs403409170	39883260	2.507 × 10^−7^	G/A	*CDCA2*/exone 12
	2	rs417405143	92030810	2.507 × 10^−7^	C/A	*ENSOARG00000014477*/96,187 bp
	2	rs412292790	93575956	2.507 × 10^−7^	A/G	Intergenic variant/286,027 bp
	2	rs398401406	93587069	2.507 × 10^−7^	G/A	Intergenic variant/297,140 bp
	12	rs421040859	4244267	2.507 × 10^−7^	A/G	*C4BPA*/intron 9–10
	20	rs414911966	31825656	4.628 × 10^−9^	G/T	*RIPOR2*/intron 1–2
WC	16	rs428638112	19304437	2.881 × 10^−7^	A/G	*ENSOARG00000007198*/intron 3–4
	16	rs415643604	37754935	3.321 × 10^−7^	G/T	*ENSOARG00000026965* (LincRNA)/Intron 1–2
	18	rs414923885	19954441	3.059 × 10^−7^	T/C	*ENSOARG00000026436* (LincRNA)/Intron 1–2
	26	rs429375956	1405366	6.893 × 10^−8^	G/A	*ENSOARG00000026782* (LincRNA)/Intron 1–2
	26	rs418752484	1415599	6.893 × 10^−8^	C/T	*ENSOARG00000026782* (LincRNA)/Intron 1–2
	26	rs429422002	12501336	1.247 × 10^−7^	G/T	*TENM3*/intron 21–22
WME	X	rs161648030	94700436	1.778 × 10^−7^	C/T	*RTL8A*/11,982 bp
	X	rs427877945	94700506	1.778 × 10^−7^	T/C	*RTL8A*/11,912 bp
	X	rs413531430	94965602	4.384 × 10^−7^	C/T	*MOSPD1*/39,398 bp
	X	rs415654848	94674867	9.063 × 10^−7^	G/A	*RTL8C*/3′ downstream
WB	9	rs406848373	73358324	9.809 × 10^−7^	A/G	*RIMS2*/intron 4–5
	15	rs427196452	51838676	3.803 × 10^−7^	G/T	*P4HA3*/intron 6–7

Chr: chromosome; DC: depth of chest; WC: width of chest; WME: width of muscle «eye»; WB: width of back.

**Table 3 genes-13-01337-t003:** Body measurements of Jalgin merino with different genotypes.

	Trait	Genotype A, n = 30, M ± m	Genotype B, n = 15, M ± m	*p*-Value
1.	Live weight at birth, kg	4.54 ± 0.07	4.41 ± 0.09	0.264
2.	Live weight at 12 months, kg	51.28 ± 0.77	47.87 ± 0.99	0.009
3.	Daily weight gain, kg	0.121 ± 0.002	0.112 ± 0.003	0.009
4.	Height at wither, cm	70.33 ± 0.26	69.80 ± 0.34	0.211
5.	Height at croup, cm	70.50 ± 0.27	70.07 ± 0.33	0.308
6.	Width of back, cm	26.00 ± 0.28	26.40 ± 0.49	0.476
7.	Width of chest, cm	24.06 ± 0.18	24.80 ± 0.27	0.029
8.	Depth of chest, cm	34.43 ± 0.18	34.13 ± 0.32	0.397
9.	Girth of shoulder, cm	31.56 ± 0.19	31.47 ± 0.22	0.729
10.	Girth of forearm, cm	20.27 ± 0.13	20.20 ± 0.21	0.780
11.	Girth of thigh, cm	34.96 ± 0.10	35.20 ± 0.15	0.197
12.	Depth of muscle «eye», mm	19.55 ± 0.07	20.03 ± 0.09	0.0003
13.	Width of muscle «eye», mm	38.16 ± 0.38	41.86 ± 0.49	0.000001
14.	Depth of adipose tissue, mm	3.42 ± 0.07	3.52 ± 0.12	0.515
15.	Depth of femoral muscle, mm	124.55 ± 0.68	123.21 ± 0.91	0.235

## Data Availability

Data are available from the corresponding author upon request.

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
