# Peer review of "A Genome-Wide Search for Candidate Genes of Meat Production in Jalgin Merino Considering Known Productivity Genes"

_genes, 2022, doi:10.3390/genes13081337_

Round 1

Reviewer 1 Report

This is an interesting paper that describes new candidate genes associated with meat production parameters.

However, I suggest that the manuscript needs to be improved before consideration for publication.

1. The title of the manuscript should be shortened.

2. The photo of Jalgin merino could be given as a Figure 1.

3. In Manhattan plots of GWAS results (Figure 2) the SNPs with unknown positions (Chromosome 0) should be removed. The authors should describe if this is correct to perform GWAS in X chromosome in rams along with autosomes.  

4. Were the rams selected for the study genetically tested for breed purity? The authors can use genome-wide SNP analysis to compare the selected rams of Jalgin merino with some other merino breeds (e.g. Stavropol, Australian). PCA and/or Admixture analysis could be given as a Figure in the manuscript.

Author Response

Dear Reviewer,

I accept all recommended changes. Thank you very much for your work, it will improve our manuscript for much better.

Regards,

Alexander Krivoruchko, corresponding author.

Reviewer 1.

This is an interesting paper that describes new candidate genes associated with meat production parameters.

However, I suggest that the manuscript needs to be improved before consideration for publication.

  1. The title of the manuscript should be shortened.

Changed to “GENOME-WIDE SEARCH FOR CANDIDATE GENES OF MEAT PRODUCTION IN JALGIN MERINO CONSIDERING OF KNOWN PRODUCTIVITY GENES”

  1. The photo of Jalgin merino could be given as a Figure 1.

Photo of Jalgin merino added in Figure 1. Figure 1. Ram (A) and ewe (B) of Jalgin merino.

  1. In Manhattan plots of GWAS results (Figure 2) the SNPs with unknown positions (Chromosome 0) should be removed.

 SNPs with unknown positions (Chromosome 0) were removed from plots. In Materials and Metods added “after removing  432 SNPs with unknown positions (Chromosome 0), 560 381 SNPs used for GWAS.”

The authors should describe if this is correct to perform GWAS in X chromosome in rams along with autosomes.

When associations are detected, each SNP is tested individually and this result does not affect the associations of other SNPs located both on the same and on different chromosomes. Therefore, we did not exclude polymorphisms located on chromosome X from testing. In addition, the Plink software takes into account the sex of animals and uses correct algorithms when assessing the quality of genotyping (including Hardy-Weinberg calculations) and associations of polymorphisms on the X chromosome. To recognize the presence of a sex chromosome during the test, the --ovine command is also added. In Materials and Metods added “SNPs on the X chromosome were not excluded, since the Plink software allows to search for associations in males, taking this into account also when controlling the quality of genotyping.”

  1. Were the rams selected for the study genetically tested for breed purity? The authors can use genome-wide SNP analysis to compare the selected rams of Jalgin merino with some other merino breeds (e.g. Stavropol, Australian). PCA and/or Admixture analysis could be given as a Figure in the manuscript.

Added in Methods: “To confirm the purity of the breed, we used the genotyping data obtained by us for 50 rams of the Manych Merino breed and 50 rams of the Stavropol breed”

“Principal component analysis were performed using the “SNPRelate” package in the programming language “R“”

Added in Results: “Figure 2 Breed purity PCA of Jalgin merino (DjalM) compared to Manych merino (ManM) and Stavropol sheep (Stav).”

“To confirm the purity of the Jalgin merino breed, a comparative analysis of genotypes was performed with the two main breeds most common in the Stavropol Territory - the Manych Merino and the Stavropol breed (Figure 2). In the space of the first two principal components, these breeds are clearly clustered into three groups, without significant intersections with each other.”

Added in Discussion: “The analysis of genotypes based on the method of principal components confirmed the purity of our sample of animals of the Jalgin merino breed. At the same time, two Merino breeds (Jalgin and Manych merinos) cluster among themselves only in the second component, which indicates fairly close origin. The Stavropol breed is clustered completely separately in both components, being located in the first component between the Merino breeds. This is consistent with the history of the origin of the studied merinos, since the sheep of the Stavropol breed were among their common ancestors.”

Reviewer 2 Report

The article is interesting, however, it lacks to include data of the analyses that were carried out, also the style of the quotations is not adequate for the magazine, check that it is written in the style of the magazine.

1.-Title is long.

2.- Materials and methods.

Used 224 animals that met the meat quality parameters. "problem case". But they failed to include a group of animals, as negative control, which are animals with low meat quality. What are the parameters for each genotype?.How many animals of each genotype were used?

More specifics, to indicate how many milliliters of blood they used, how much dna they obtained and in which laboratory they sent it for analysis.

missing: Analysis of Gene Ontology and Metabolic Pathways.with significant snp, to identify the metabolic pathways that are affected.

3.- Result.

After QC analysis, how many SNPs were genotyped?

how many SNP, were excluded for not passing the HWE and MAF test?

How many SNPs were used to carry out the GWAS test?.

Can validate presence of SNPs in genes (MSTN, MEF2B, FABP4), in both groups of genotypes, by amplifying the region where the snp is located, by PCR?.

Author Response

Dear Reviewer,

I accept all recommended changes. Thank you very much for your work, it will improve our manuscript for much better.

Regards,

Alexander Krivoruchko, corresponding author.

Reviewer 2.

The article is interesting, however, it lacks to include data of the analyses that were carried out, also the style of the quotations is not adequate for the magazine, check that it is written in the style of the magazine.

We are improved the style of the quotations in text.

Need to improve English style

After correction paper will be send to MDPI Proof-Reading service.

1.-Title is long.

Changed to “GENOME-WIDE SEARCH FOR CANDIDATE GENES OF MEAT PRODUCTION IN JALGIN MERINO CONSIDERING OF KNOWN PRODUCTIVITY GENES”

2.- Materials and methods.

Used 224 animals that met the meat quality parameters. "problem case". But they failed to include a group of animals, as negative control, which are animals with low meat quality. What are the parameters for each genotype?.How many animals of each genotype were used?

When searching for associations, we did not divide animals into case-control groups based on high or low meat productivity. For the analysis, the method of quantitative associations was used (command --qassoc in Plink). The software ranked the animals for each of the lifetime indicators of meat productivity and performed an associative analysis with the presence or absence of SNPs. In the group of examined animals, there was a dispersion of the phenotype according to the selected productivity parameters, which made it possible to study the association of quantitative parameters with the genotype. Already on the basis of the identified SNPs with a high association with productivity parameters, we divided the animals into two groups: genotype “A” and genotype “B”, which included 30 and 15 individuals, respectively. For them, an analysis of differences in the parameters of the phenotype was carried out, shown in Table 3. The analysis once again confirmed the relationship of the polymorphisms we identified with parameters of meat productivity of sheep.

More specifics, to indicate how many milliliters of blood they used, how much dna they obtained and in which laboratory they sent it for analysis.

Added “DNA was isolated from 100 µl of blood, the average concentration was 25 ng/µl.” and “Genotyping was carried out in the laboratory of the Skolkovo Institute of Science and Technology SCOLTECH (Moscow, Russian Federation)”

missing: Analysis of Gene Ontology and Metabolic Pathways.with significant snp, to identify the metabolic pathways that are affected.

Dear reviewer, gene ontology analysis was not part of our research plans. However, on your recommendation, we have supplemented this section. Thanks for the recommendation, it will make our article better.

Added in Materials and Methods:

“Gene ontology (GO) analysis was performed using the g:Profiler (version e106_eg53_p16_65fcd97, database updated on 18/05/2022). Sets of more than 5 and less than 500 genes were selected for testing. The results were considered significant at p<0.05.”

Added in Results:

“Figure 6. Gene ontology (GO) analysis for discovered candidate genes in Jalgin merino.”

“In our study, the results of a genome-wide search for candidate genes for productive traits were supplemented by an ontology analysis of genes and metabolic pathways that may affect the manifestation of phenotypic traits. As a result, it was found that out of 14 potential candidate genes in GO analysis, three Linc RNAs and two uncharacterized sequences did not pass. For the remaining candidate genes, 15 biological processes and 5 molecular functions were significantly identified (p<0.05) (Figure 6). Molecular functions have been associated with proline metabolism and ascorbic acid transport. Biological processes included regulatory pathways for the development and operation of neurons, morphogenesis of sensory organs, and regulation of neutrophil functions. A number of metabolic pathways have been associated with protein metabolism, guanyl biotransformation and GTP.”

Added in Discussion:

“From the five molecular functions, discovered during GO analysis, four are associated with the activity of proline dioxygenase, also described as prolyl hydroxylase. This enzyme is widely represented in various organs and tissues, as it takes part in the modification of one of the most common connective tissue proteins, collagen. With its participation, the correct tertiary structure of the protein is formed and its functional activity is ensured (Gorres 2010). The function associated with the transport of ascorbic acid provides the body with a natural antioxidant that prevents damage to proteins and DNA. In addition, ascorbic acid is an important cofactor in the synthesis of collagen and cholesterol, and is involved in the metabolism of xenobiotics. Sufficient intake of ascorbic acid in the animal's body is necessary for its normal growth and development (Matsui 2012). Taking into account the important role of collagen and ascorbic acid in the growth and development of animals, we consider the involvement of the candidate genes identified by us in the implementation of this molecular function as confirmation of their influence on the productive qualities of the Jalgin merino. Among biological processes, many are associated with the development and functions of neurons, including the morphogenesis of sensory organs. Thus, they ensure the normal functioning of the nervous system of the animal, which is also responsible for the formation of normal eating behavior. It depends on providing the body with enough nutrients necessary for growth and development (Nishijo 2020). A number of biological processes include the regulation of guanyl-nucleotide exchange factor activity and GTP binding. They are united by their influence on the transmembrane transport of nutrients and ensuring the response of the cell to external influences (Wu 2011). Biological processes associated with the functions of neutrophils are part of the normal functioning of the immune system. The resistance of the body to infections and parasitic invasions ensures the normal level of metabolism of animals necessary for their growth and development (Guzman 2018). Thus, the pathways we have identified in one way or another affect the phenotype of animals and take part in the realization of productive qualities in the Jalgin merino.”

Added in Rederence:

Gorres, K.L.; Raines, R.T. Prolyl 4-hydroxylase. Critical Review of Biochemical Molecular Biology. 2010, 45(2), 106-24.

Matsui, T. Vitamin C nutrition in cattle. Asian-Australians Journal of Animal Science. 2012, 25(5), 597-605.

Nishijo, H.; Ono, T. 'Neural Mechanisms of Feeding Behavior and Its Disorders', in A. Takada (ed.), New Insights Into Metabolic Syndrome, IntechOpen, 2020, London.

Wu, X.; Bradley, M.J.; Cai, Y.; Kummel, D.; De La Cruz, E.M.; Barr, F.; Reinisch, K.M. Insights regarding guanine nucleotide exchange from the structure of a DENN-domain protein complexed with its Rab GTPase substrate. Proc. Natl. Acad. Sci. USA 2011, 108, 18672–18677.

Guzman, E.; Montoya, M. Contributions of Farm Animals to Immunology. Frontiers of Veterinary Sciences. 2018, 5, 307.

3.- Result.

After QC analysis, how many SNPs were genotyped?

560 381 were genotyped after quality control, added in 2.4. Quality control of genotyping.

how many SNP, were excluded for not passing the HWE and MAF test?

39 811 SNPs failed frequency test, 834 markers were excluded based on HWE test. 12 804 SNPs failed missingness tested. All information added in 2.4. Quality control of genotyping.

How many SNPs were used to carry out the GWAS test?.

after removing  432 SNPs with unknown positions (Chromosome 0), 560 381 SNPs used for GWAS. Added in 2.4. Quality control of genotyping

Can validate presence of SNPs in genes (MSTN, MEF2B, FABP4), in both groups of genotypes, by amplifying the region where the snp is located, by PCR?.

No, we did not use PCR to confirm the presence of identified SNPs in the region of the described genes. These SNPs were identified by genotyping using the Ovine Infinium HD BeadChip 600K and we considered these results to be significant and do not need to be confirmed by additional methods.

do not use last names in citations, instead, use numbering as indicated in the journal's instructor

All citations was corrected in text.

Line 53 cites? the citation is missing, which substantiates the paragraph

The paragraph was rearranged and citation added to right position.

Line 56 the citation is missing, which substantiates the paragraph

Added citation “Grochowska, Ewa, Borys, Bronislav, Mroczkowski, Slavomir. (2019) Effects of Intronic SNPs in the Myostatin Gene on Growth and Carcass Traits in Colored Polish Merino Sheep. Genes, 11(1):2. doi: 10.3390/genes11010002.”

Line 67 cites? paragraph without citation

Added citation: “Passamonti, Matilde Maria, Somenzi, Elisa, Barbato, Mario, Chillemi, Giovanni, Colli, Licia, Joost, Stéphane, Milanesi, Marco, Negrini, Riccardo, Santini, Monia, Vajana, Elia, Williams, John Lewis, and Ajmone-Marsan, Paolo. (2021) The Quest for Genes Involved in Adaptation to Climate Change in Ruminant Livestock. Animals (Basel). 11(10):2833. doi: 10.3390/ani11102833.”

Line 92 used 224 animals that met the meat quality parameters. "problem case". But they failed to include a group of animals, as negative control, which are animals with low meat quality.

When searching for associations, we did not divide animals into case-control groups based on high or low meat productivity. For the analysis, the method of quantitative associations was used (command --qassoc in Plink). The software ranked the animals for each of the lifetime indicators of meat productivity and performed an associative analysis with the presence or absence of SNPs. In the group of examined animals, there was a dispersion of the phenotype according to the selected productivity parameters, which made it possible to study the association of quantitative parameters with the genotype.

Line 101 any reference, where the same or similar method was used?

Added two citations:

Brito, Luiz F., Clarke, Shannon M., McEwan, John C., Miller, Stephen P., Pickering, Natalie K., Bain, Wendy E., Dodds, Ken G., Sargolzaei, Mehdi, and Schenkel, Flávio S. (2017) Prediction of genomic breeding values for growth, carcass and meat quality traits in a multi-breed sheep population using a HD SNP chip. BMC Genetics, 18, 7. https://doi.org/10.1186/s12863-017-0476-8

Zhang, Li, Liu, Jiasen, Zhao, Fuping, Ren, Hangxing, Xu, Lingyang, Lu, Jian, Zhang, Shifang, Zhang, Xiaoning, Wei, Caihong, Lu, Guobin, Zheng, Youmin, Du, Lixin. (2013) Genome-wide association studies for growth and meat production traits in sheep. PLoS One, 8(6):e66569. doi: 10.1371/journal.pone.0066569.

Line 106 more specifics, to indicate how many milliliters of blood they used, how much dna they obtained and in which laboratory they sent it for analysis.

Added “DNA was isolated from 100 µl of blood, the average concentration was 25 ng/µl.” and “Genotyping was carried out in the laboratory of the Skolkovo Institute of Science and Technology SCOLTECH (Moscow, Russian Federation)”

Line 129 The association map and the significant SNPs were not plotted.

We presented the data of the associations of individual SNPs with the studied phenotype parameters in the form of Table 2. The first column presents the studied phenotype parameters, and the third column groups the SNPs associated with them, ranked by chromosomes and positions on the chromosomes. The following are quantitative characteristics of the significance of the identified associations. Since the main goal of the work was to search for new candidate genes for productivity, we focused more on identifying the SNPs located next to the ones we found and with a high probability of genes inherited together with them. They are listed in the last column of the table.

Line 366 Выявленные полиморфизмы находились в экзонах, интронах и 366 прилегающих областях следующих генов: CDCA2, ENSOARG00000014477, C4BPA, RI- 367 POR2, ENSOARG00000007198.

Changed to “The identified polymorphisms were found in the exons, introns and adjacent regions of the following genes and transcripts: CDCA2, ENSOARG00000014477, C4BPA, RIPOR2, ENSOARG00000007198,…”

Round 2

Reviewer 1 Report

The authors have addressed all the suggestions I made in my review. The quality of the article has been improved.

Author Response

Dear Reviewer,

Thank you very much for recommendation.   Best regards, Alexander Krivoruchko, corresponding author

Reviewer 2 Report

The manuscript lacks greater structure. Authors are recommended to review other publications on the same subject, such as doi:10.3390/ani10030434, to adequately express the results and the discussion.

English is confused and not fluent. It is recommended to write the manuscript with experts in the English language.

line 51: c.*1232G>A, what is the meaning of the character "*"

line 89: Change section 2.1. Ethics statement by 2.1. Animals Used and Obtaining Samples. they must explain the characteristics of the animals, indicate the coordinates of the sites, the collection of samples, also in this section include Ethics statement, insert figures 1 and 2.

Line 107: section 2.3 genotyping change to 2.3 Extraction of DNA and Genotyping.

line: 147: It is not correct to start with the results, mentioning the objectives. It should start by indicating the Genotyping Analysis and Data Quality Control

line 149: figures 1 and fig 2, you must relocate it in section 2.1

line 147:the results must start describing the values Genotyping Analysis and Data Quality Control.

Line 260: change figure 6, it is not a figure, it is a table. I suggest plotting the values of  Gene ontology categories identified in genes with significant SNPs.

Author Response

Dear Reviewer,

Thank you very much for recommendation, we accept all of them in our manuscript.  

The manuscript lacks greater structure. Authors are recommended to review other publications on the same subject, such as doi:10.3390/ani10030434, to adequately express the results and the discussion.

Thank you for recommendation. We analyzed article and add some data and reference in our manuscript.

English is confused and not fluent. It is recommended to write the manuscript with experts in the English language.

The manuscript was passed MDPI proofreading (you can see changes highlighted in text).

line 51: c.*1232G>A, what is the meaning of the character "*"

This is parameter of HGVS nomenclature. It is commented at http://varnomen.hgvs.org/recommendations/general/

It mean: “*” (asterisk) is used in nucleotide numbering and to indicate a translation termination (stop) codon (see Standards); c.*32G>A and p.Trp41*

line 89: Change section 2.1. Ethics statement by 2.1. Animals Used and Obtaining Samples. they must explain the characteristics of the animals, indicate the coordinates of the sites, the collection of samples, also in this section include Ethics statement, insert figures 1 and 2.

Corrected based on your recommendations. Section 2.1. named “Animals Used and Obtaining Samples”, figures 1 and 2 were inserted.

Line 107: section 2.3 genotyping change to 2.3 Extraction of DNA and Genotyping.

Section 2.3 changed to “2.2 Extraction of DNA and Genotyping”

line: 147: It is not correct to start with the results, mentioning the objectives. It should start by indicating the Genotyping Analysis and Data Quality Control

Start of results corrected on your recommendations.

line 149: figures 1 and fig 2, you must relocate it in section 2.1

Figures are relocated.

line 147:the results must start describing the values Genotyping Analysis and Data Quality Control.

Start of results corrected on your recommendations.

Line 260: change figure 6, it is not a figure, it is a table. I suggest plotting the values of  Gene ontology categories identified in genes with significant SNPs.

Figure 6 was edited. In figure added values of Gene ontology categories like in article your recommended as good quality primer.

Best regards,

Alexander Krivoruchko, corresponding author.